# Structure of bacterial cytoplasmic chemoreceptor arrays and implications for chemotactic signaling

Ariane Briegel[1], Mark S Ladinsky[1], Catherine Oikonomou[2], Christopher W Jones[3], Michael J Harris[4], Daniel J Fowler[4†], Yi-Wei Chang[2], Lynmarie K Thompson[4], Judith P Armitage[3], Grant J Jensen[1,2]*

[1]Division of Biology and Biological Engineering, California Institute of Technology, Pasadena, United States; [2]Howard Hughes Medical Institute, Pasadena, United States; [3]Department of Biochemistry, University of Oxford, Oxford, United Kingdom; [4]Department of Chemistry, University of Massachusetts, Amherst, United States

**Abstract** Most motile bacteria sense and respond to their environment through a transmembrane chemoreceptor array whose structure and function have been well-studied, but many species also contain an additional cluster of chemoreceptors in their cytoplasm. Although the cytoplasmic cluster is essential for normal chemotaxis in some organisms, its structure and function remain unknown. Here we use electron cryotomography to image the cytoplasmic chemoreceptor cluster in *Rhodobacter sphaeroides* and *Vibrio cholerae*. We show that just like transmembrane arrays, cytoplasmic clusters contain trimers-of-receptor-dimers organized in 12-nm hexagonal arrays. In contrast to transmembrane arrays, however, cytoplasmic clusters comprise two CheA/CheW baseplates sandwiching two opposed receptor arrays. We further show that cytoplasmic fragments of normally transmembrane *E. coli* chemoreceptors form similar sandwiched structures in the presence of molecular crowding agents. Together these results suggest that the 12-nm hexagonal architecture is fundamentally important and that sandwiching and crowding can replace the stabilizing effect of the membrane.

*For correspondence: jensen@caltech.edu

Present address: †Department of Molecular Physiology and Biophysics, University of Vermont, Burlington, United States

**Competing interests:** The authors declare that no competing interests exist.

**Reviewing editor**: Jodi Nunnari, University of California, Davis, United States

## Introduction

Bacteria sense and respond to their environment through a chemotactic system that translates ligand binding (stimulus) into preferential flagellar rotation (response), leading to either running in a favorable direction or tumbling to find a more favorable direction (*Hazelbauer et al., 2008*). Briefly, ligands bind to the periplasmic domains of the methyl-accepting chemotaxis proteins (MCPs), either directly (*Milburn et al., 1991*; *Englert et al., 2010*) or via periplasmic binding proteins (*Tam and Saier, 1993*). This results in conformational changes that traverse the length of the MCP, through one or more HAMP (histidine kinases, adenyl cyclases, MCPs, and some phosphatases) domains and the cytoplasmic coiled-coil signaling domain to either activate or inactivate a histidine kinase, CheA, bound at the distal tip (*Wadhams and Armitage, 2004*; *Kentner and Sourjik, 2006*; *Hazelbauer et al., 2008*).

The bacterial chemotactic system is best understood in *Escherichia coli*, where system components are located in a single operon (*Silverman and Simon, 1976*). In *E. coli*, CheA is a 5-domain (P1-P5) protein that functions as a homodimer. When activated by the MCP in unfavorable environments, CheA undergoes autophosphorylation and then transfers the phosphoryl group to one of two response regulators. One of these, CheY, binds the flagellar motor when phosphorylated, switching the motor's direction from clockwise to counter-clockwise and thus increasing the frequency of tumbling to find a

**eLife digest** Many bacteria swim through water by rotating tiny hair-like structures called flagella. In *E. coli*, if all the flagella on the surface of a bacterium rotate in a counterclockwise fashion, then it will swim in a particular direction, but if the flagella all rotate in an clockwise fashion, then the bacterium will stop swimming and start to tumble.

Bacteria use a combination of swimming and tumbling in order to move towards or away from certain chemicals. For example, a bacterium is able to move towards a source of nutrients because it is constantly evaluating its environment and will swim forward for longer periods of time when it recognizes the concentration of the nutrient is increasing. And if it senses that the nutrient concentration is decreasing, it will tumble in an effort to move in a different direction.

Many bacteria, such as *E. coli*, rely on proteins in their cell membrane called chemoreceptors to sense specific chemicals and then send signals that tell the flagella how to rotate. These transmembrane receptors and their role in chemotaxis—that is, movement towards or away from specific chemicals in the environment—have been widely studied. However, other bacteria also have chemoreceptors in the cytoplasm inside the bacterial cell, and much less is known about these.

Now, Briegel et al. have examined the cytoplasmic chemoreceptors of two unrelated bacteria, *R. sphaeroides* and *V. cholera*, and found that the cytoplasmic chemoreceptors arrange themselves in hexagonal arrays, similar to the way that transmembrane chemoreceptors are arranged. However, the cytoplasmic chemoreceptors arrange themselves in a two-layer sandwich-like structure, whereas the transmembrane chemoreceptors are arranged in just one layer. The next step is to understand how chemical binding causes these arrays to send their signals to the motor. A complete understanding of this signaling system may ultimately allow scientists to re-engineer it to draw bacteria to targets of medical or environmental interest, such as cancer cells or contaminated soils.

more favorable direction (*Turner et al., 2000*). The other response regulator, CheB, is a methylesterase whose activity is stimulated by phosphorylation. Its activity is opposed by the constitutively active methyltransferase CheR and the balance of these two activities determines the methylation state of specific glutamate residues in the MCPs. These methylations confer adaptation on the system, modulating its response based on recent environmental conditions (*Kleene et al., 1979*; *Toews et al., 1979*; *Lupas and Stock, 1989*).

In *E. coli*, MCPs associate into trimers of dimers that span the inner membrane (*Kim et al., 1999*; *Studdert and Parkinson, 2004*) and complex with CheA and the coupling protein CheW (*Figure 1A*). This results in hexagonally packed trimers-of-receptor-dimers surrounding a ring of alternating CheA P5 domains and CheW. Neighboring rings are linked together by CheA P3 dimerization domains to form extended hexagonal lattices – the so-called transmembrane chemoreceptor arrays (*Briegel et al., 2012*; *Liu et al., 2012*). The architecture of these membrane-bound chemoreceptor arrays is not specific to *E. coli*, indeed it is universal among bacteria (*Briegel et al., 2009*). The architecture of these arrays is likely key to the high sensitivity, wide dynamic range, cooperativity, and feedback control of this system (*Duke and Bray, 1999*; *Li and Weis, 2000*; *Gestwicki and Kiessling, 2002*; *Sourjik and Berg, 2002*, *2004*; *Li and Hazelbauer, 2005*; *Endres and Wingreen, 2006*).

Besides the single chemosensory operon encoding components of the transmembrane array, exemplified by *E. coli*, many bacteria contain additional chemotaxis operons, whose products are less well understood. The best-studied case is *Rhodobacter sphaeroides*, in which there are three operons, two of which are essential for normal chemotaxis in laboratory conditions. One of these operons, CheOp$_2$, encodes components of the chemotaxis array associated with the transmembrane MCPs. The other, CheOp$_3$, encodes components of a cytoplasmic chemotaxis cluster including CheB, CheR, CheY, CheW, two atypical CheA proteins (CheA$_3$ and CheA$_4$) (*Porter et al., 2008*), and a soluble chemoreceptor, TlpT. Each of the CheA proteins in the cluster lacks canonical domains compared to *E. coli* CheA: CheA$_3$ contains only the P1 and P5 domains separated by a 794 amino acid sequence of unknown structure, whereas CheA$_4$ lacks the P1 and P2 domains. CheA$_4$ can transfer phosphoryl groups to P1 of CheA$_3$ (*Porter et al., 2008*). Both CheA$_3$ and CheA$_4$ are required for chemotaxis (*Wadhams et al., 2003*; *Porter et al., 2008*). A second MCP homolog lacking any transmembrane domain, TlpC, is

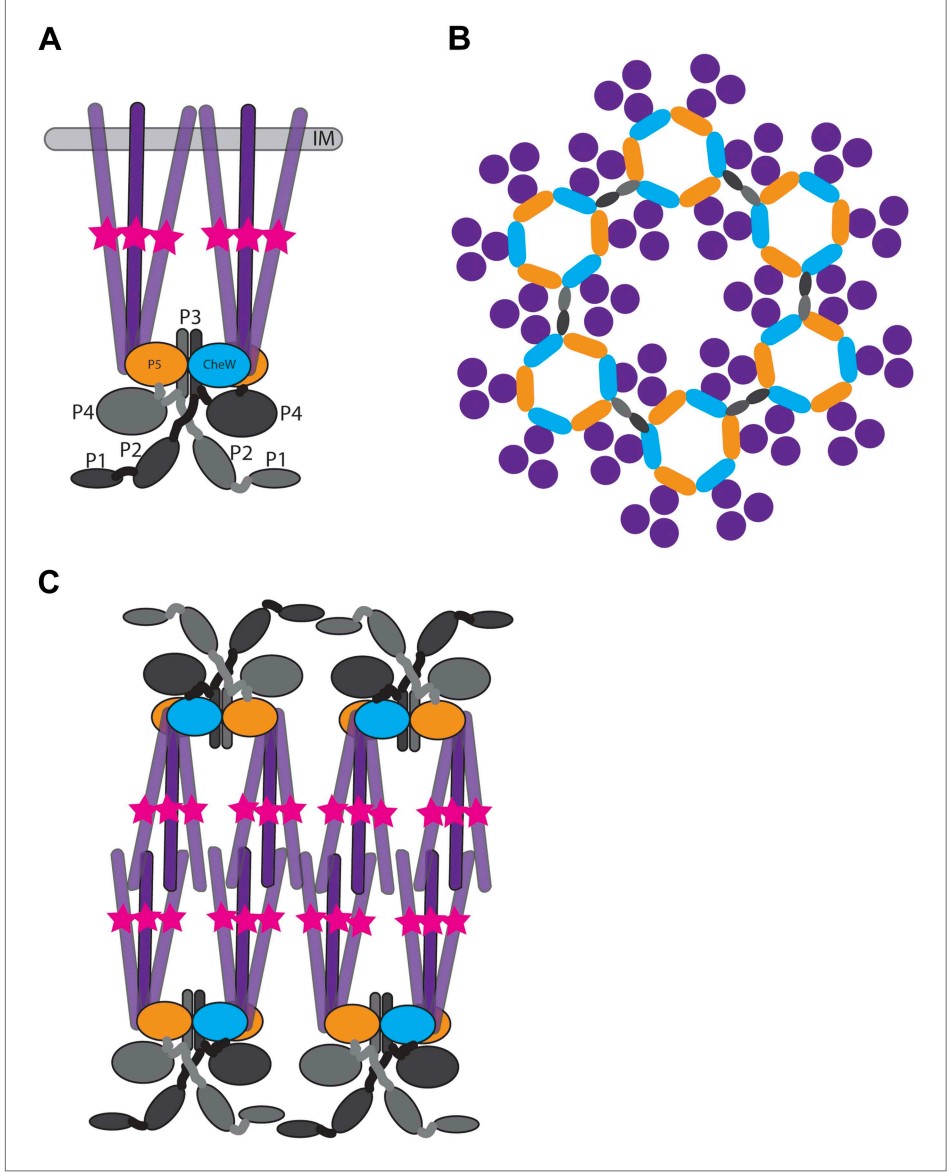

**Figure 1**. Structure of membrane-bound and cytoplasmic receptor complexes. (**A**) Schematic showing the topology of receptor-trimers-of-dimers (purple), CheA (domains P1–P4 light gray/dark gray indicating domains in each CheA monomer, domain P5 orange), and CheW (blue) in membrane-bound arrays. The methylation region of each receptor dimer is indicated by a pink star. IM = inner membrane. (**B**) Top-view of the arrangement of the array components showing the interaction sites between the receptors, CheA and CheW, colored as in (**A**). (**C**) Schematic showing the topology of receptor, CheA, and CheW complexes in cytoplasmic arrays, colored as in (**A**). Cytoplasmic chemoreceptors assemble into two hexagonally packed arrays interacting at their presumably ligand-binding tips. DOI: 10.7554/eLife.02151.003

encoded in $CheOp_2$. Fluorescent fusions of both TlpT and TlpC localize to a single cytoplasmic focus at mid-cell, which divides and is segregated during cell division (*Wadhams et al., 2002*, *2003*). $CheA_3$ and $CheA_4$, along with $CheW_4$ and a CheR, also localize to this cytoplasmic cluster (*Wadhams et al., 2003*). $CheW_4$ and TlpT are required to form the cluster, and the positioning of the cluster depends on the ParA homolog PpfA, which is encoded next to TlpT in $CheOp_3$ (*Thompson et al., 2006*). PpfA is a chromosome-associated ParA-like ATPase and controls the localization and segregation of the cytoplasmic cluster through an interaction with the N-terminus of TlpT, which is thought to stimulate the ATPase activity of PpfA in a ParB-like manner (*Roberts et al., 2012*). Sequence analysis of TlpT suggests that the 567 amino acid protein forms a classical MCP coiled-coil structure (*Alexander and Zhulin, 2007*).

The 580 amino acid TlpC lacks significant homology to the canonical MCP structure and lacks a recognizable HAMP domain or methylation region.

Cytoplasmic chemoreceptors are not unique to *R. sphaeroides*. The genomes of many bacterial species encode MCP homologs lacking transmembrane domains (*Ulrich and Zhulin, 2010*; *Wuichet and Zhulin, 2010*). However, the organization and precise function of these cytoplasmic chemoreceptor clusters remain unknown. In the case of *R. sphaeroides*, it has been speculated that cytoplasmic MCPs modulate chemotactic response based on the current metabolic state of the cell (*Armitage and Schmitt, 1997*; *Porter et al., 2008*), while the inputs of cytoplasmic receptors in other organisms such as *Vibrio cholerae* are less clear.

In the case of *E. coli* transmembrane arrays, the membrane is thought to be important for proper assembly and function (*Miller and Falke, 2004*; *Draheim et al., 2006*; *Amin and Hazelbauer, 2012*). We were therefore interested in how cytoplasmic receptors cluster in the absence of organizing membrane. We used tomography of freeze-substituted, plastic-embedded sections; immunoelectron microscopy; electron cryotomography (ECT) of both intact cells and cryosections (*Gan and Jensen, 2012*); and correlated cryogenic fluorescence light microscopy and ECT (cryo-FLM/ECT) to characterize the structure of cytoplasmic chemoreceptor clusters in two bacteria: *R. sphaeroides* and *V. cholerae*. We also report the ability of normally transmembrane chemoreceptors from *E. coli* to form cytoplasmic-like arrays in the absence of a membrane.

## Results

### *R. sphaeroides* cells overexpressing cytoplasmic chemoreceptor array components exhibit large, curved structures

We initially tried to image cytoplasmic chemoreceptor arrays in intact *R. sphaeroides by* ECT. Unfortunately, we found that the size and cytoplasmic density of this species limits the achievable resolution in tomograms of intact cells, and we could not resolve the cytoplasmic clusters. To circumvent this problem, we employed two approaches that allowed us to view sections of the otherwise thick samples: cryosectioning vitreously frozen cells and room temperature sectioning of high-pressure frozen, freeze-substituted cells.

Tomographic reconstructions of cryosections of *R. sphaeroides* cells (containing the chemotaxis protein TlpC tagged with GFP to aid in future identification) revealed curved, double-layered structures that might correspond to cytoplasmic clusters (*Figure 2A,B*), but they were small and difficult to differentiate from the complex environment. To gain confidence in our identification, we used a strain in which some cytoplasmic chemotaxis components are overexpressed. This strain exhibited chemotaxis in swim plate assays, indicating that the limited overexpression did not significantly impair chemotactic function. By fluorescence light microscopy (FLM), short cells contained single fluorescent foci and long cells two foci, as expected (*Wadhams et al., 2002*). In tomograms of cryosections of these cells, we observed similar structures to those seen in WT cells but with greater length, appearing in cross-section as a curved, double-layered sandwich approximately 27 nm wide (measured from density peak-to-peak; *Figure 2C–E*).

It was difficult to identify the location of the structure in the cryosections, so we turned to a technique that would allow us to place the structure in the context of the whole cell. In tomograms of high-pressure frozen, freeze-substituted cells, we again observed extended, double-layered structures (*Figure 3*). Cross-sections through the structures revealed two curved layers with perpendicular striations connecting them. The positions of the structures in the cells were consistent with FLM observations (*Thompson et al., 2006*), strengthening our confidence in the identification of the cytoplasmic array. We did occasionally see multiple clusters at these positions, consistent with fluorescence results in cells overexpressing these proteins.

### Immunogold labeling and correlated FLM/ECT confirm structures are cytoplasmic chemoreceptor arrays

To check whether the structures we observed contained the cytoplasmic chemotaxis protein TlpC, we performed immunogold labeling of the GFP tag in our sections. In this technique, gold-conjugated anti-GFP antibodies recognize and bind epitopes on the surface of the section, which is subsequently imaged by tomography. We consistently observed gold labeling of the curved structures described above, and tomography of the volume beneath the label confirmed the structural details seen in cryosections

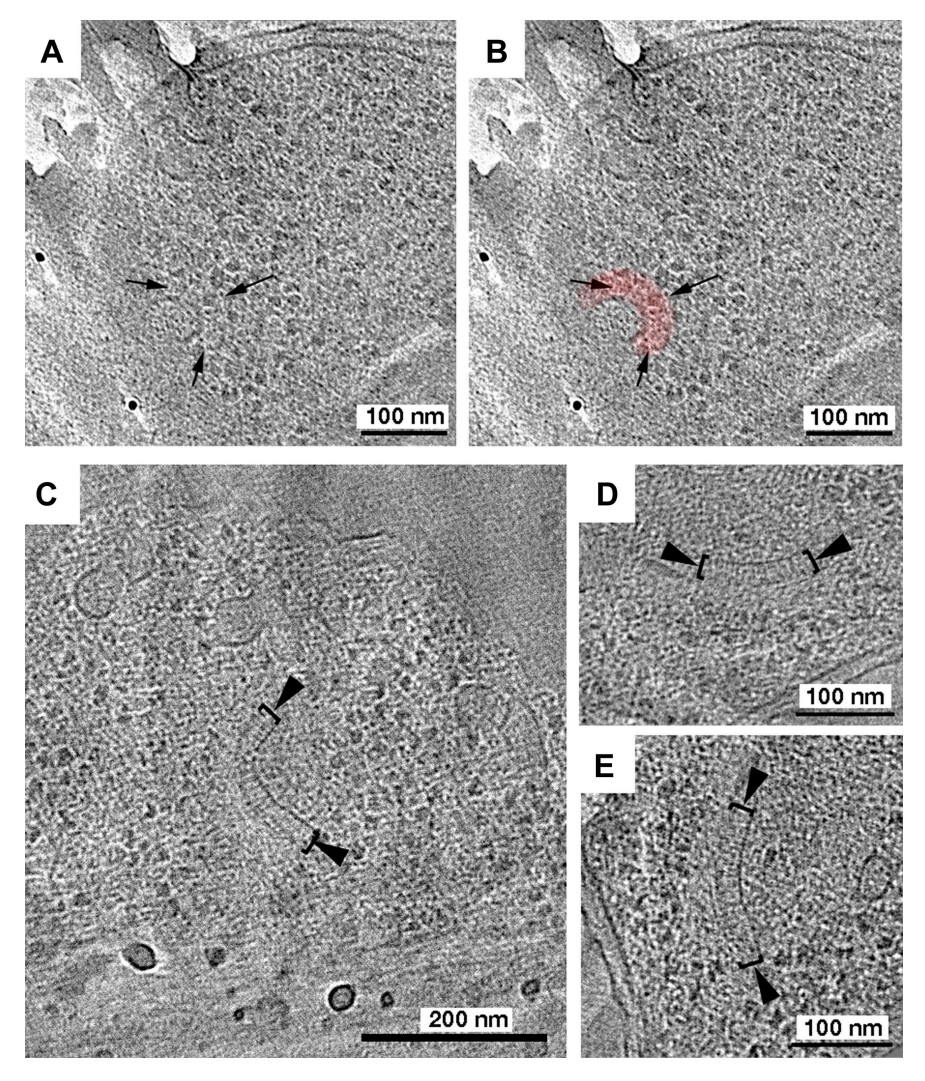

**Figure 2**. Tomography of *R. sphaeroides* cryosections reveals cytoplasmic clusters. Tomographic slices of cryosections through *R. sphaeroides* cells expressing TlpC-GFP at wild-type (**A** and **B**) or overexpressed (**C**–**E**) levels. (**A**) Tomographic slice of a cell expressing TlpC-GFP, revealing a potential cytoplasmic chemoreceptor array (black arrows), pseudo-colored red in (**B**). (**C**–**E**) Cryosections of cells overexpressing cytoplasmic chemoreceptor components, including TlpC-GFP, contain similar structures to those observed under WT expression conditions: two curved layers approximately 27 nm apart (brackets), with perpendicular striations between them (black arrowheads).

(n = 17 cells, example shown in *Figure 4*), indicating that these structures are indeed cytoplasmic chemo-receptor arrays.

To corroborate this identification, we used correlated cryo-FLM/ECT. Since whole *R. sphaeroides* cells are too thick and dense to image by ECT with high resolution, we flattened them by gentle lysis. However, unlike transmembrane receptors that remain well-organized following cell lysis (*Briegel et al., 2009*, *2012*, *2013*), fluorescent foci were not observed after lysis, indicating disruption of the cytoplasmic receptor cluster. We reasoned that in the absence of membrane, the density of the cytoplasm might play an important role in stabilizing cytoplasmic clusters. We therefore added molecular crowding agents, either polyethylene glycol (PEG-8000) or polyvinylpirrolidone (PVP), to the lysis protocol and used this approach to image an *R. sphaeroides* strain overexpressing the cytoplasmic chemore-ceptor TlpT fused to YFP. We observed preservation of TlpT-YFP signals by FLM after lysis (*Figure 5A–C*). We then froze lysed cells on an EM grid and localized a single TlpT-YFP signal by cryo-FLM, then

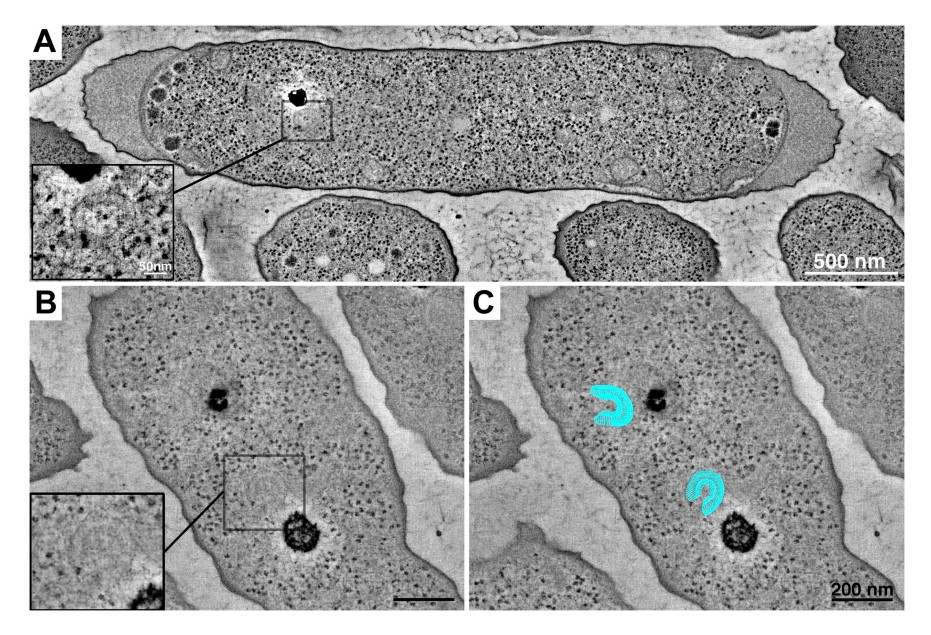

**Figure 3**. Tomography of cytoplasmic clusters in freeze-substituted *R. sphaeroides*. Dual-axis tomographic reconstruction of high-pressure frozen, freeze-substituted cells reveals cytoplasmic clusters in most cells. Clusters appear as partially circular structures, typically located near storage granules. (**A**) Reconstruction of a longitudinally sectioned cell showing a single cluster. Inset: high-magnification detail of the cluster in a single, 6.7-nm tomographic slice. Scale bar 50 nm. (**B**) Cell containing two cytoplasmic clusters located next to storage granules. Inset: high-magnification detail of one cluster. (**C**) Model (blue) superimposed over the tomogram shown in **B**, highlighting the clusters. Scale bars 200 nm.

imaged the indicated location at high resolution by ECT. We observed that the fluorescence signal corresponded to the structure previously identified (*Figure 5D–F*), further verifying that these are indeed cytoplasmic chemoreceptor arrays containing TlpT.

## Hexagonal receptor arrangement is conserved between cytoplasmic and transmembrane arrays

We compared the structure of the cytoplasmic chemoreceptor cluster identified by correlated cryo-FLM/ECT to that of a transmembrane receptor cluster in the same cell. Interestingly, the arrangement and packing of the receptors was identical in both cases: a hexagonal lattice with center-to-center spacing of 12 nm when viewed from above (*Figure 5G–J*). In the transmembrane array, this receptor lattice is associated with a single baseplate, whereas in the cytoplasmic array, there are two, one on either side, approximately 30 nm apart. Unfortunately, due to the tight curvature, we could only confirm the hexagonal order of one of the two layers of the cytoplasmic array.

## ECT of *V. cholerae* reveals double-baseplate architecture of cytoplasmic arrays

As discussed above, due to the substantial density and thickness of *R. sphaeroides* cells, we were unable to identify cytoplasmic chemoreceptor arrays in intact cells. To obtain higher resolution information from intact cells, we turned to *V. cholerae*, which is also predicted to contain cytoplasmic chemoreceptors and flattens significantly during freezing for cryo-EM, yielding a thinner sample. By ECT of whole *V. cholerae* cells, we observed cytoplasmic chemoreceptor clusters with the same architecture as those in *R. sphaeroides*, though lacking curvature (*Figure 6*). The improved resolution in the thinner cells allowed us to clearly distinguish two hexagonal arrays of trimers-of-receptor-dimers sandwiched between two baseplates, 35 nm apart (*Figure 6*). The kinase-binding regions of the receptors were highly ordered, with decreasing order toward the center, as previously observed for transmembrane chemoreceptor arrays.

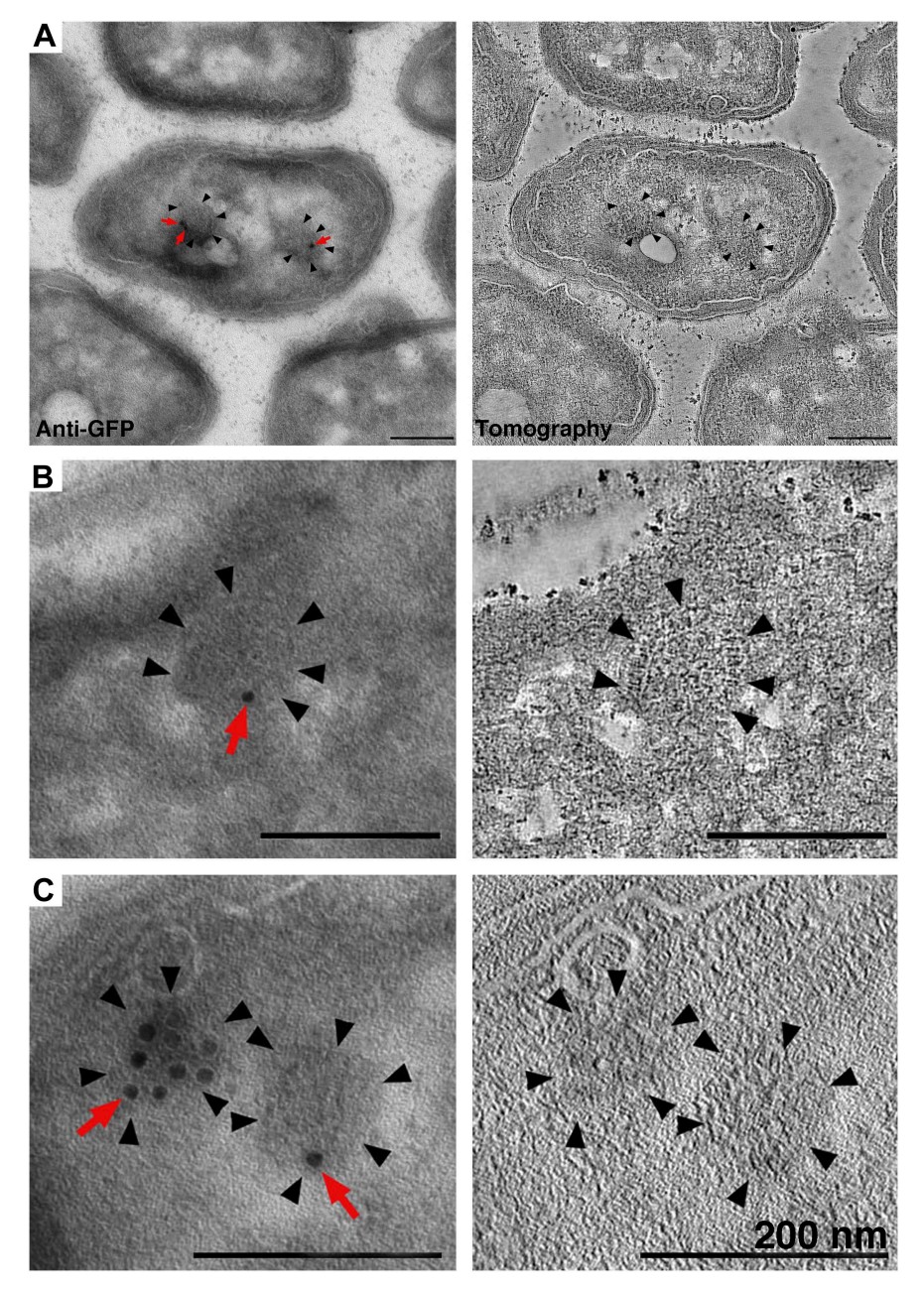

**Figure 4**. Identification of cytoplasmic clusters in *R. sphaeroides* by tomography of immunolabeled, negatively stained sections. Sections were immunolabeled with antibodies against GFP and gold-conjugated secondary antibodies. (**A**–**C**) Three examples of immunolabeled cells (left) and corresponding tomographic slices (right). Clusters (black arrowheads) were typically labeled by 1–3 gold particles (red arrows), depending on the amount of antigen present on the section surface. The heavily labeled cluster in (**C**) is likely oriented *en face* and near the section surface, allowing for access to a larger number of GFP antigens. Scale bars 200 nm.

## Transmembrane *E. coli* chemoreceptors can form cytoplasmic-like arrays in the absence of membrane

We observed a striking similarity between the cytoplasmic arrays in *R. sphaeroides* and *V. cholerae* and an in vitro chemoreceptor preparation from *E. coli*. We purified a cytoplasmic fragment of the aspartate-sensing Tar receptor, as well as CheA and CheW, from *E. coli* (*Montefusco et al., 2007*;

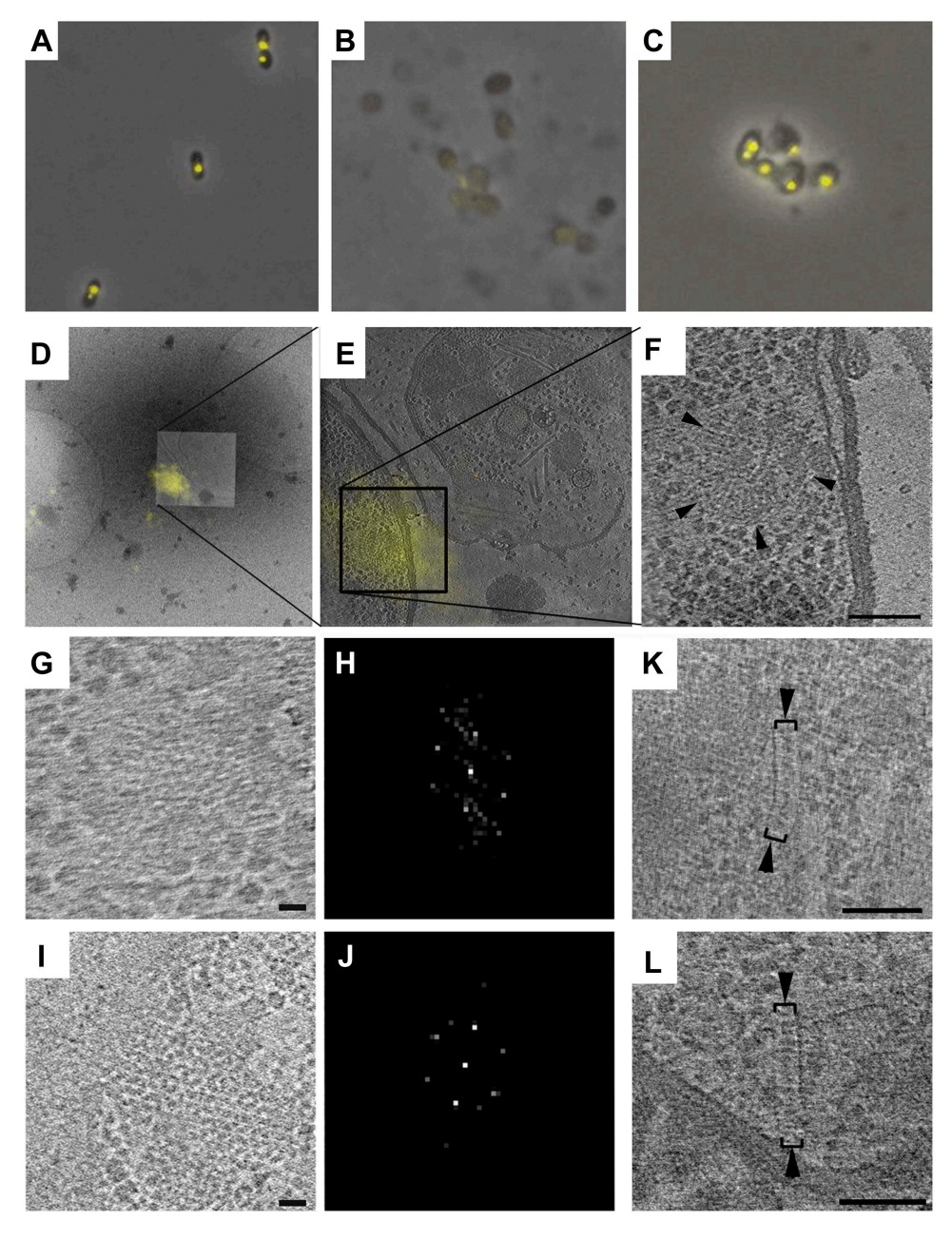

**Figure 5**. Correlative cryo-FLM/ECT of *R. spheroides* cytoplasmic arrays. (**A–C**) Molecular crowding preserves cytoplasmic chemoreceptor clusters during lysis. Overlay of phase contrast and TlpT-YFP fluorescence images of *R. sphaeroides* cells of strain JPA1558OE before (**A**) and after lysis either in the absence (**B**) or presence (**C**) of 10% PVP. (**D–F**) Correlative cryo-FLM/ECT identifies a cytoplasmic array. (**D**) Overlay of fluorescent signal from TlpT-YFP (yellow) and a low magnification cryo-EM image of the same lysed cell. (**E**) Overlay of TlpT-YFP fluorescent signal (yellow) and a cross-section of a reconstructed tomogram corresponding to the same region, identifying the structure of a cytoplasmic chemoreceptor array. (**F**) An enlarged view of the cytoplasmic array reveals a partial ring consisting of two layers of chemoreceptors. Scale bar 100 nm. (**G–J**) Receptor packing is identical in *R. sphaeroides* cytoplasmic and transmembrane arrays. (**G**) Top view through one of the two layers of the array imaged in **D–F** and corresponding power spectrum (**H**) reveal a hexagonal receptor arrangement with center-to-center spacing of 12 nm. Scale bar 25 nm; power spectrum not to scale. (**I**) Top view through a membrane-bound chemoreceptor array and corresponding power spectrum (**J**) reveal hexagonal packing identical to that of the cytoplasmic array shown in (**G–H**). Scale bar 25 nm; power spectrum not to scale. (**K–L**) Additional examples of cytoplasmic chemoreceptor

*Figure 5. Continued on next page*

*Figure 5. Continued*
arrays (brackets with arrowheads) in tomograms of *R. sphaeroides* cells lysed with molecular crowding agents added. Distance between baseplates is approximately 30 nm. Scale bars 100 nm.

*Fowler et al., 2010*) and assembled complexes in vitro using these components, with CheA and CheW present in excess. The mixture of purified proteins also contained the molecular crowding agents PEG-8000 and trehalose to simulate cytoplasmic conditions. Once formed, the complexes activated the kinase as effectively as when assembled on vesicles (*Fowler et al., 2010*). Using ECT, we observed extended arrays with identical architecture to that of the in vivo cytoplasmic clusters described above: two baseplates, approximately 31 nm apart, flank two hexagonal lattices of chemoreceptor trimers with 12 nm center-to-center spacing (*Figure 7*). The receptors from the two sides interact at their methylation tips. Again, as we observed for transmembrane chemoreceptors, the kinase-binding tips were well ordered, with decreasing order towards the middle of the sandwich.

## Curvature of *R. sphaeroides* cytoplasmic arrays is an intrinsic property

The most striking difference between the cytoplasmic arrays in *R. sphaeroides* and *V. cholerae* was the curvature observed in the former. In all cases observed, cytoplasmic *V. cholerae* arrays and arrays formed in vitro from *E. coli* cytoplasmic receptor fragments were flat, while those of *R. sphaeroides* exhibited kinks and regions of high curvature, even forming full rings in some cases. We wondered if this curvature may be induced by interactions between the fluorophores rather than being a property of the arrays themselves, as tags are known to introduce such artifacts (e.g., *Swulius and Jensen, 2012*). To test this, we performed tomography of high-pressure frozen, freeze-substituted cells overexpressing

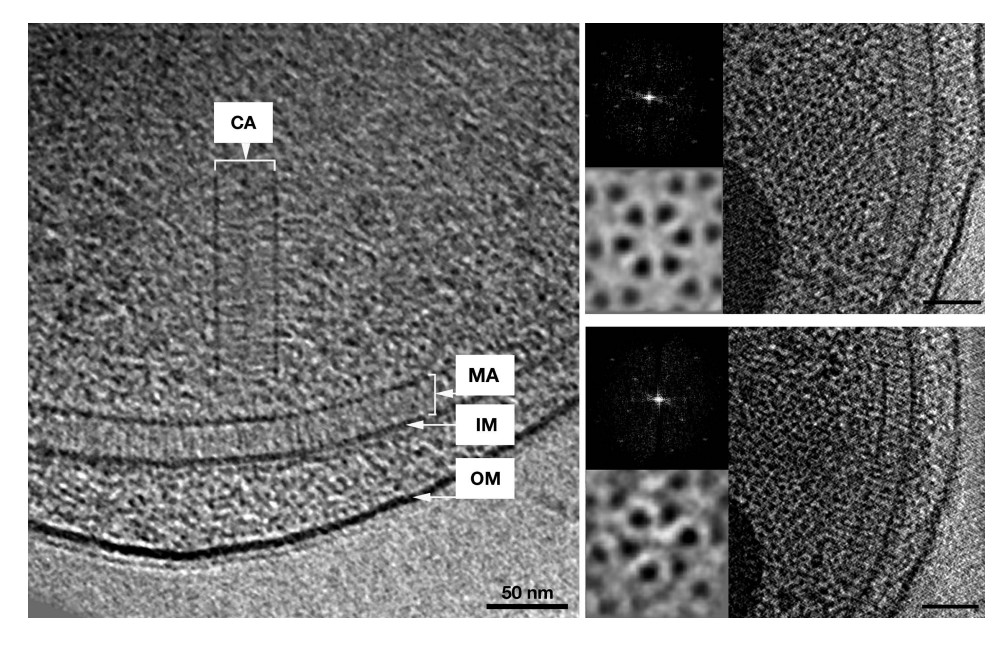

**Figure 6**. In vivo architecture of the *V. cholerae* cytoplasmic array. Left: side view of a membrane-bound chemoreceptor array (MA) and a cytoplasmic chemoreceptor array (CA). The cytoplasmic array is composed of two parallel CheA/W baseplates approximately 35 nm apart. The chemoreceptors are sandwiched between the two baseplates and are hexagonally packed with a 12 nm center-to-center spacing. Right: top views of the receptor packing close to the CheA/CheW baseplate on either side and corresponding power spectra (top insets), as well as sixfold symmetrized subvolume averages (bottom insets) reveal that the hexagonal arrangement of the receptors is identical to that of the membrane bound array described previously (*Briegel et al., 2009*). Scale bars 50 nm. CA, Cytoplasmic chemoreceptor array; MA, membrane-bound chemoreceptor array; IM, inner membrane; OM, outer membrane. Power spectra not to scale.

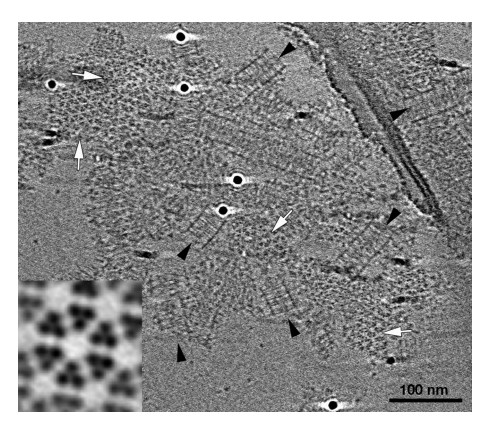

**Figure 7**. *E. coli* Tar chemoreceptors lacking transmembrane regions can form cytoplasmic-like arrays in the presence of CheA, CheW, and a molecular crowding agent. Tomographic slice showing cytoplasmic fragments of Tar forming extended arrays in the presence of CheA, CheW, and molecular crowding agents. These arrays closely resemble the cytoplasmic chemoreceptor arrays seen in *V. cholerae* (*Figure 6*). Side-view (black arrowheads) reveals two flat, parallel CheA/W baseplates spaced approximately 31 nm apart. Top views of the chemoreceptors close to the baseplates (white arrows) reveal a well-ordered, hexagonal arrangement with a center-to-center spacing of 12 nm. Enlarged subvolume average (inset) confirms that the packing is identical to that of in vivo arrays. Scale bar 100 nm.

untagged components of the cytoplasmic cluster. We observed structures identical to those observed in the tagged strains (*Figure 8A*), indicating that curvature is not an artifact induced by attached fluorophores.

Cytoplasmic arrays in *R. sphaeroides* are segregated by a ParA homolog, PpfA, which associates with the N-terminus of TlpT and with the chromosomal DNA. To test whether this interaction imposes curvature on the cytoplasmic array, we similarly imaged cells lacking PpfA. Again, we observed structures with similar curvature, indicating that this is an intrinsic property of the array, not affected by the segregation machinery (*Figure 8B*).

## Discussion

Over half of all genomes sequenced of motile bacteria have more than one putative chemosensory pathway and many of these encode putative soluble chemoreceptors (*Wuichet and Zhulin, 2010*). In this study, we describe the structure of cytoplasmic chemoreceptor arrays in two distantly related bacterial species, *R. sphaeroides* and *V. cholerae*. Notably, cytoplasmic arrays in both species display a hexagonal arrangement of receptor-trimers-of-dimers, with 12 nm center-to-center spacing that is identical to that of transmembrane chemoreceptor arrays, suggesting a fundamental utility for this architecture. Whereas the transmembrane array is embedded in the membrane with a CheA/W baseplate at the receptors' membrane-distal tips and ligand-binding domains in the periplasm, the cytoplasmic arrays assemble as a sandwich, with two CheA/W baseplates flanking two sheets of MCPs that interact in the middle, presumably at their ligand-binding domains (*Figure 1C*). No membrane is associated with these arrays. The identity of the outer baseplates as CheA/W is supported by our observation of the

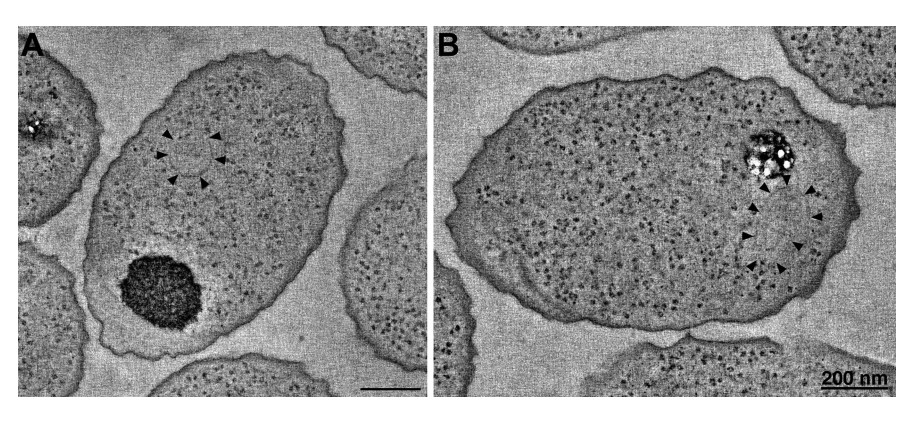

**Figure 8**. *R. sphaeroides* cytoplasmic arrays are inherently curved. (**A**) Tomographic slice of a high-pressure frozen, freeze-substituted cell overexpressing native, untagged components of the cytoplasmic chemoreceptor cluster. Arrowheads indicate curved array. (**B**) Tomographic slice of a similarly treated cell lacking the partitioning protein PpfA. Arrowheads indicate curved cytoplasmic array. Scale bars 200 nm.

same structure in in vitro preparations from *E. coli*, where only three proteins are present: receptor fragments, CheA, and CheW. The similarity of the array organization in membrane-bound and cytoplasmic clusters suggests a common organizational theme, in which either the membrane or the crowded environment of the cytoplasm provides necessary stability to the conserved architecture of a 12 nm hexagonal array of receptor-trimers-of-dimers.

Interestingly, while transmembrane arrays remain intact upon cell lysis, presumably due to the stabilizing effect of the membrane, cytoplasmic arrays fall apart unless molecular crowding agents are added. This suggests that cellular crowding contributes to the stability of the array and may be an important factor in assembly of these, and likely other, cellular structures (*Ellis, 2001*; *Zhou et al., 2008*; *Zhou, 2013*). We observe the same requirement for molecular crowding (mimicking the cellular environment) in our in vitro preparation from *E. coli*. This supports the idea that chemoreceptor interactions can be enhanced by membrane binding (*Shrout et al., 2003*), but in the absence of a membrane, this can be achieved by molecular crowding and sandwiching (*Fowler et al., 2010*).

The hexagonal order of the *R. sphaeroides* and *V. cholerae* cytoplasmic arrays, with 12 nm center-to-center spacing, is identical to the arrangement of transmembrane chemoreceptor arrays in other bacteria (*Briegel et al., 2009*), as well as to arrays formed from cytoplasmic fragments of *E. coli* receptors. This indicates that the atypical *R. sphaeroides* CheA homologs (CheA$_3$ and CheA$_4$), despite their sequence anomalies, assemble in a similar fashion to canonical CheA proteins, including the three CheA homologs found in *V. cholerae* (*Ulrich and Zhulin, 2010*). From what is known in other organisms, *R. sphaeroides* CheA$_3$ and CheA$_4$ likely intercalate into CheA/W rings with their P5 domains (*Briegel et al., 2012*; *Liu et al., 2012*; *Li et al., 2013*; *Natale et al., 2013*). This packing would position both atypical CheA homologs in close proximity to each other, thereby facilitating phosphotransfer from the P4 domain of CheA$_4$ to the P1 domain of CheA$_3$.

The most striking difference between cytoplasmic chemoreceptor arrays in *R. sphaeroides* and *V. cholerae* is their curvature. In our observations, all cytoplasmic *V. cholerae* arrays and arrays formed in vitro from *E. coli* cytoplasmic receptor fragments were flat, while those of *R. sphaeroides* exhibited kinks and regions of high curvature, even forming full rings in some cases. This indicates that curvature may not affect performance unduly, with functioning arrays found in linear, concave, and convex conformations. However, there may be an upper limit of curvature tolerance. The discontinuities we observe in cytoplasmic arrays may result from 'breaking' of the CheA/W baseplate on one side under conditions of extreme curvature.

In transmembrane arrays, ligand-binding domains are located in the periplasm. In cytoplasmic arrays, we assume that these domains are located in the middle of the sandwich. This configuration is unlikely to inhibit access to signaling molecules as the receptor arrangement is much more porous than the outer membrane, which extracellular ligands need to cross in order to bind transmembrane arrays. The signaling implications of this arrangement remain unclear, but it is intriguing to think that ligand binding in the middle of the sandwich could result in CheA activation in both baseplates, facilitating amplification.

## Materials and methods

### Strains and growth conditions

Strains used in this study are listed in *Table 1*. *R. sphaeroides* strains were cultured in succinate medium (*Sistrom, 1960*) supplemented with 25 µg/ml kanamycin and 30 µM IPTG (to induce expression of plasmid-borne FliA from pIND [*Ind et al., 2009*]) at 30°C with shaking. *V. cholerae* strains were cultured in LB media at 37°C with shaking. *V. cholerae* cells were then mixed 1:1 with another bacterial species grown in Ca-HEPES buffer (25 mM HEPES, 2 mM CaCl$_2$ at pH 7.6) and grown for an additional 16 hr at 30°C with shaking.

### Light microscopy

Cells were immobilized on agarose pads and imaged using a Nikon Eclipse 90i microscope (Nikon Instruments Inc., Melville, NY) using a 100× oil immersion lens and images were recorded on a Coolsnap HQ$^2$ camera (Photometrics, Tuscon, AZ) operated using the Metamorph software (Molecular Devices, Chicago, IL).

### Vitreous cryosectioning

Vitreous cryosections were prepared as previously described (*Ladinsky et al., 2006*; *Ladinsky, 2010*). Briefly, *R. sphaeroides* cells were pelleted and resuspended in ~100 µl of 50% dextran in SUX buffer then

**Table 1.** Strains used in this study

| Strain | Relevant genotype | Source |
|---|---|---|
| JPA543 | WS8N TlpC-GFP | *Wadhams et al., 2002* |
| JPA543OE | TlpC-GFP pIND-FliA | *This study* |
| JPA1558OE | TlpT-YFP pIND-FliA | *This study* |
| WS8NOE | pIND-FliA | *This study* |
| JPA1330OE | ΔppfA TlpC-GFP pIND-FliA | *This study* |
| MKW1383 | N16961 ΔctxAB::kan | Matthew Waldor |

rapidly-frozen in a BalTec HPM-010 high-pressure freezer (Leica Microsystems, Vienna) using dome-shaped brass planchettes, then stored in liquid nitrogen. Once opened under liquid nitrogen, the sample-containing planchette was placed in a FC6/UC6 cryoultramicrotome equipped with a model M micromanipulator (Leica Microsystems). Blockfaces were trimmed with a diamond trimming knife and sections were cut with a 25° Cryo-Platform knife (Diatome-US) at −145°C. Ribbons of sections (130 nm) were transferred to carbon-coated, 200-mesh copper grids (Electron Microscopy Sciences) and stored in liquid nitrogen.

## Electron cryotomography

20 µl cell culture was mixed with pelleted 100 µl colloidal gold solution, BSA treated to avoid aggregation (*Iancu et al., 2007*). 3 µl of this cell-gold mixture was then applied to R2/2 copper Quantifoil grids (Quantifoil Micro Tools). After blotting away excess liquid using a Vitrobot (FEI), the sample was plunge-frozen in a liquid ethane-propane mixture (*Iancu et al., 2007*; *Tivol et al., 2008*). Images were collected using either an FEI G2 (FEI Company, Hillsboro, OR) 300 kV field emission gun electron microscope or an FEI TITAN Krios (FEI Company, Hillsboro, OR) 300 kV field emission gun with an image corrector for lens aberration correction. Both microscopes were equipped with Gatan image filters (Gatan, Pleasanton, CA) and K2 Summit counting electron detector cameras (Gatan, Pleasanton, CA). Data were collected using the UCSFtomo software (*Zheng et al., 2007*) using cumulative electron doses of ~160 e/A$^2$ or less for each individual tilt-series. The images were CTF corrected, aligned, and reconstructed using weighted back projection using the IMOD software package (*Kremer et al., 1996*). SIRT reconstructions were calculated using TOMO3D (*Agulleiro and Fernandez, 2011*). Subvolume averaging and symmetrizing were done using PEET (*Nicastro et al., 2006*).

## High-pressure freezing and freeze-substitution

*R. sphaeroides* cells were grown at 30°C to stationary phase in SUX buffer, centrifuged and the pellets resuspended in SUX buffer +10% Ficoll (70 kD; Sigma). The cells were centrifuged gently and the supernatant was removed. Packed cells were placed in aluminum or brass planchettes (#39201, Ted Pella, Inc.) and high-pressure frozen as described above. Frozen samples in closed planchettes were transferred under liquid nitrogen to cryogenic vials (Nalge Nunc International, Rochester NY) containing 2% OsO$_4$, 0.05% uranyl acetate in acetone. Vials were placed in an AFS freeze-substitution machine (Leica Microsystems) and processed at −90°C for 72 hr, warmed over 10 hr to −20°C and further processed at that temperature for 24 hr. The samples were brought to room temperature, rinsed four times with acetone, removed from the planchettes and infiltrated with Epon-Araldite epoxy resin (Electron Microscopy Sciences, Hatfield PA). Samples were placed in embedding molds and polymerized at 60°C for 24 hr.

## Electron tomography of plastic-embedded cells

Embedded pellets of *R. sphaeroides* cells were serially sectioned with an EM-UC6 ultramicrotome (Leica Microsystems) using a diamond knife (Diatome-US, Hatfield PA). Thick (350 nm) sections were placed on Formvar-coated, copper-rhodium 1-mm slot grids (Electron Microscopy Sciences) and stained with 3% aqueous uranyl acetate and lead citrate. Colloidal gold particles (10 nm) were placed on both surfaces of the grid to serve as fiducial markers for image alignment. Grids were placed in a model 2040 dual-axis tomography holder (Fischione Instruments, Export PA) and imaged with a Tecnai TF30ST-FEG transmission electron microscope (FEI) at 300 keV. Dual-axis tilt-series were acquired automatically using the SerialEM software package (*Mastronarde, 2005*). Samples were tilted ± 60° and imaged at 1° intervals about orthogonal axes. Images were recorded digitally with an UltraScan 994 1000XP camera (Gatan, Inc. Pleasanton CA). Tomographic data was subsequently processed, analyzed, and modeled using the IMOD software package (*Kremer et al., 1996*).

## Immunoelectron microscopy

*R. sphaeroides* cells were cultured to stationary phase as described above. TlpC-GFP signal was verified by fluorescence microscopy. The cells were pelleted, the supernatant was removed and the pellet was resuspended in 10% paraformaldehyde, 5% sucrose in SUX buffer. The cells were pelleted again and the supernatant was replaced with fresh fixative without disturbing the pellet. The cells were fixed overnight at 4°C. The samples were brought to room temperature, the fixative was removed and the pellets were infiltrated into 2.1M sucrose in PBS over the course of 1 day, with the sucrose solution changed at 1-hr intervals. The pellets were transferred to aluminum sectioning stubs and rapidly frozen in liquid nitrogen.

Semi-thin (90 nm) cryosections were cut at −110°C with a FC6/UC6 cryoultramicrotome (Leica Microsystems) using a Cryo-Immuno diamond knife (Diatome-US). Cryosections were picked up in a drop of 2.3M sucrose in PBS and transferred to Formvar-coated, carbon-coated, glow-discharged 100-mesh copper-rhodium grids. Sections were incubated for 30' with 10% calf serum in PBS to block nonspecific antibody binding sites, then labeled with a monoclonal antibody against GFP (Rockland Immunochemicals, Inc. Gilbertsville PA) diluted in 5% calf serum in PBS, followed by a colloidal gold (15 nm) conjugated anti-mouse secondary antibody (BBI International, Grand Forks ND). After labeling, the sections were negatively stained with 1% uranyl acetate and stabilized with 1% methylcellulose. Immunolabeled sections were imaged with a Tecnai T12 electron microscope at 120 keV. Tomographic tilt-series were acquired and analyzed as described above.

## *R. sphaeroides* cell lysis

Mid-log phase cells of strain JPA1558 were collected by centrifugation and resuspended in crowding buffer (20% sucrose, 7% PEG-8000, 4% trehalose, 25 mM Tris–HCl, pH 7.0) with 2 mg/ml lysozyme, then incubated for 15 min at room temperature. Following addition of 2 mM $MgCl_2$, 250 µM $CaCl_2$, and 1 mg/ml DNase I, cells were incubated for 30 min at 37°C, collected by centrifugation and resuspended in crowding buffer for imaging. Identical results were obtained with a different crowding agent, replacing the PEG-8000 and trehalose with 10% PVP and treating the cells as before.

## Correlative cryo-FLM/ECT

Cells of strain JPA1558 were lysed as described above and plunge-frozen on copper EM finder grids (Quantifoil Micro Tools), loaded into a cryo-FLM stage (FEI company) and imaged on a Nikon 90Ti inverted microscope (Nikon Instruments Inc., Melville, NY) using a 60x ELWD air objective lens and a Neo 5.5 sCMOS camera (Andor Technology, South Windsor, USA) using the NIS Elements software (Nikon Instruments Inc., Melville, NY). The grid was then transferred to the cryo-EM and tilt series were recorded from the same cells imaged by FLM previously. The grid was kept below −150°C at all times during the imaging process.

## Purification and assembly of *E. coli* chemotaxis components

Cytoplasmic fragments of the Tar receptor ($CF_{4Q}$), CheW, CheA, and CheY were expressed and purified as previously described (*Fowler et al., 2010*). The Tar fragments contained methylation-mimicking glutamine residues at the four major sites of receptor methylation. Protein purity was assessed with SDS-PAGE analysis, and protein concentrations were determined using a BCA assay (Thermo Fisher Scientific). PEG 8000 (Fluka) and D-(+)-trehalose (Sigma–Aldrich) were prepared as 40% (wt/vol) stock solutions in deionized water and passed through a 0.22-µm syringe filter prior to use. A modified kinase buffer (50 mM potassium phosphate, 50 mM KCl, 5 mM $MgCl_2$, pH 7.5) was used for sample preparation.

Formation and characterization of kinase-active ternary complexes followed published methods (*Fowler et al., 2010*; *Mudiyanselage et al., 2013*), further specified as follows. PEG-mediated $CF_{4Q}$ complexes were prepared by incubating 50 µM $CF_{4Q}$, 20 µM CheW, and 12 µM CheA with final concentrations of 7.5% wt/vol PEG 8000 and 4% wt/vol trehalose. $CF_{4Q}$ was added last to minimize CF-promoted aggregation (*Montefusco et al., 2007*) and samples were incubated overnight at 25°C in a circulating water bath and subjected to an enzyme-coupled assay and gel-based cosedimentation assay to check for phosphorylation activity and ternary complex formation. Kinase activity of PEG-mediated $CF_{4Q}$ complexes was similar to that observed for complexes assembled via binding to vesicles.

## Acknowledgements

The authors wish to thank Lee Rettberg and Audrey Huang for experimental help, and Dr Matthew Waldor for kindly providing strain MKW1383. Special thanks to Drs Zhiheng Yu and Jason de la Cruz

for microscopy support at HHMI Janelia Farms. JPA and CWJ thank BBSRC and Nikon for funding. This work was additionally supported by NIH grant R01-GM085288 to LKT and NIGMS grant GM101425 to GJJ.

## Additional information

### Funding

| Funder | Grant reference number | Author |
| --- | --- | --- |
| National Institute of General Medical Sciences | GM101425 | Ariane Briegel, Grant J Jensen |
| Howard Hughes Medical Institute | | Catherine Oikonomou, Yi-Wei Chang, Grant J Jensen |
| National Institutes of Health | R01-GM085288 | Lynmarie K Thompson |
| Biotechnology and Biological Sciences Research Council | | Christopher W Jones, Judith P Armitage |
| Nikon | | Christopher W Jones, Judith P Armitage |

The funders had no role in study design, data collection and interpretation, or the decision to submit the work for publication.

### Author contributions

AB, MSL, CO, Conception and design, Acquisition of data, Analysis and interpretation of data, Drafting or revising the article; CWJ, Conception and design, Analysis and interpretation of data, Drafting or revising the article, Contributed unpublished essential data or reagents; MJH, Acquisition of data, Analysis and interpretation of data, Drafting or revising the article, Contributed unpublished essential data or reagents; DJF, Conception and design, Contributed unpublished essential data or reagents; Y-WC, Acquisition of data, Analysis and interpretation of data, Drafting or revising the article; LKT, JPA, GJJ, Conception and design, Analysis and interpretation of data, Drafting or revising the article

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
