## [Decision Letter]

Thank you for sending your work entitled “Structure of bacterial cytoplasmic chemoreceptor arrays and implications for chemotactic signaling” for consideration at *eLife*. Your article has been favorably evaluated by a Senior editor, a Reviewing editor, and 3 reviewers, one of whom is a member of our Board of Reviewing Editors.

The Reviewing editor and the reviewers discussed their comments before we reached this decision, and the Reviewing editor has summarized the critical concerns that will need to be addressed in a revised manuscript.

The consensus is that the structural analysis of bacterial chemoreceptor arrays in the manuscript represents a significant advance in understanding bacterial chemosensing and chemotaxis mechanism. However, the main conclusions of the manuscript would be significantly strengthened by experiments directed at imaging native *R. sphaeroides* arrays and/or confirming the identity of the cholera structures by immuno-EM. The reviewers have also requested that you revise the text of your manuscript to include previous electron cryo-tomography work on bacterial chemotaxis receptor arrays with associated citations. In addition, there are several excellent suggestions made by Reviewer 3 for organizational and text revisions that would improve the clarity of the manuscript.

*Reviewer #2*:

This is a well conceived and written study that represents a fundamental new contribution to the field of bacterial chemosensing and chemotaxis. The ECT structural characterization of cytoplasmic chemotaxis arrays is novel and highly significant. Certainly the work is worthy of publication in a top journal.

This reviewer can see only two significant issues worth considering:

1) The MS suggests that the planar organization of polar and cytoplasmic complexes is dominated by the CheA/W base plate. However, the fact that naked receptors form hexagonal arrays (zippers) suggests that the receptors have an intrinsic propensity to form hexagonal arrays, perhaps by providing hydrophobic binding sites arranged on the outside of the trimer with the proper symmetry for hexagonal assemblies. It is likely an oversimplification to say that the array organization is provided primarily by the base plate – the structure of the receptor trimer of dimers is essential too.

2) If possible, it would be nice to include images of native *R. sphaeroides* arrays, in order to confirm the hypothesis that the observed curvature of the GFP-fusion arrays is due to the non-native GFP component. Perhaps these native arrays could be visualized by FLM/ECT in the lysed cell system using a non-hydrolyzable, fluorescent ATP derivative or an anti-Che antibody. However, this could be beyond the scope of the current MS.

*Reviewer #3*:

1) The authors describe the architecture of cytosolic chemoreceptors in *R. sphaeroides*, and also present some structural data from the cytosolic chemoreceptor of *V. cholera* in vivo and purified *E. coli* Tar chemoreceptors lacking transmembrane regions. Interestingly, the authors show that these structures, previously demonstrated to localize to discrete foci in cells, all share a common back-to-back architecture of hexagonally packed arrays. They propose that the (periplasmic, in membrane-bound arrays) ligand binding domains are thus minimally exposed to the cytosol. Through the finding that soluble domains of *E. coli* receptors form a similar structure, the authors argue that it may represent a default, low-energy configuration of chemoreceptor arrays rather than a specific evolutionary adaptation. However, the biological significance of this architecture is not discussed. Moreover, the interesting finding that the *R. sphaeroides* structure is curved is clouded by the lack of data from wild-type cells – the authors acknowledge this curvature may be due to the fluorescent proteins fusions.

2) The initial description of receptor topology is somewhat confusing. Would a diagram be helpful? Some detailed information about the domains may not be necessary.

3) What new information does Figure 2 add? It is helpful to have an orthogonal technique, but perhaps this could be combined with Figure 1.

4) The claim is made, based on Figure 2, that the receptors are localized to the 1/4 and 3/4 positions. This should be backed up by quantification, and the mentioned similarity to light microscopy appropriately cited.

5) One novel aspect of the *R. sphaeroides* chemoreceptor arrays is that they appear to be curved (though, as already mentioned, the exact nature of this curvature was not clear to me until later in the manuscript). The authors mention that the observed curvature may be due to the fluorescent protein tag. Imaging WT cells would obviously resolve this issue and would strengthen the manuscript considerably. While data they present from other species suggests that the formation of arrays can occur without a FP tag, the possibility that array formation is mediated by this tag remains.

6) What is the biological significance of arrays in general, and the cytoplasmic ones in particular? While large portions are devoted to structural details, the functional significance of these details is lacking.

---

## [Author Response]

*The consensus is that the structural analysis of bacterial chemoreceptor arrays in the manuscript represents a significant advance in understanding bacterial chemosensing and chemotaxis mechanism. However, the main conclusions of the manuscript would be significantly strengthened by experiments directed at imaging native* R. sphaeroides *arrays and/or confirming the identity of the cholera structures by immuno-EM. The reviewers have also requested that you revise the text of your manuscript to include previous electron cryo-tomography work on bacterial chemotaxis receptor arrays with associated citations. In addition, there are several excellent suggestions made by reviewer 3 for organizational and text revisions that would improve the clarity of the manuscript*.

We wish to thank both the editors and the reviewers for excellent comments and suggestions on our manuscript.

Imaging native *R. sphaeroides* arrays is very challenging – the structures are very rare and only part of each cell is captured in a cryosection. We did, however, share the reviewers’ concern that the fluorescent tags we used to facilitate identification by correlative cryo-FLM/ECT could introduce artifacts. Therefore, we have added tomography of *R. sphaeroides* overexpressing native, untagged chemotaxis components (new Figure 8). The cytoplasmic arrays observed in this strain were indistinguishable from those imaged previously, showing that the curvature of the array is not an artifact of fluorophore-tagging.

We didn’t confirm the identity of the *V. cholera* structures by immuno-EM because we don’t know which of the multiple chemoreceptors lacking predicted transmembrane regions in the genome are found in the clusters, and we don’t have an appropriate antibody. However, we feel that it is unnecessary due to the striking conservation of the array architecture with trimer-of-receptor- dimers at the vertices of a 12 nm hexagonal lattice, a conformation that is universally conserved among bacterial transmembrane arrays (7). Also, we observe an identical arrangement of two hexagonally packed arrays sandwiched between CheA/CheW baseplates in the complexes formed by normally transmembrane *E. coli* receptor fragments.

We have included all relevant citations to our previous ECT work on chemoreceptor arrays. We have incorporated the text revisions suggested by Reviewer 3.

In addition, we have also added another experiment demonstrating that the ParA homolog PpfA, responsible for segregating the cytoplasmic clusters, does not induce curvature of the *R. sphaeroides* arrays.

Reviewer #2:

*This is a well conceived and written study that represents a fundamental new contribution to the field of bacterial chemosensing and chemotaxis. The ECT structural characterization of cytoplasmic chemotaxis arrays is novel and highly significant. Certainly the work is worthy of publication in a top journal*.

*This reviewer can see only two significant issues worth considering*:

*1) The MS suggests that the planar organization of polar and cytoplasmic complexes is dominated by the CheA/W base plate. However, the fact that naked receptors form hexagonal arrays (zippers) suggests that the receptors have an intrinsic propensity to form hexagonal arrays, perhaps by providing hydrophobic binding sites arranged on the outside of the trimer with the proper symmetry for hexagonal assemblies. It is likely an oversimplification to say that the array organization is provided primarily by the base plate* – *the structure of the receptor trimer of dimers is essential too*.

Thank you. We agree that the basic structure of trimers-of-receptor-dimers is fundamental and we have included the following sentence in the text: “The similarity of the array organization in membrane-bound and cytoplasmic clusters suggests a common organizational theme, in which either the membrane or the crowded environment of the cytoplasm provides necessary stability to the conserved architecture of a 12 nm hexagonal array of receptor-trimers-of-dimers.”

*2) If possible, it would be nice to include images of native* R. sphaeroides *arrays, in order to confirm the hypothesis that the observed curvature of the GFP-fusion arrays is due to the non-native GFP component. Perhaps these native arrays could be visualized by FLM/ECT in the lysed cell system using a non-hydrolyzable, fluorescent ATP derivative or an anti-Che antibody. However, this could be beyond the scope of the current MS*.

We have included electron tomography of untagged *R. sphaeroides* arrays, confirming that the curvature of the array is not an artifact induced by fluorophore tagging.

Reviewer #3:

*1) The authors describe the architecture of cytosolic chemoreceptors in* R. sphaeroides*, and also present some structural data from the cytosolic chemoreceptor of* V. cholera in vivo *and purified* E. coli *Tar chemoreceptors lacking transmembrane regions. Interestingly, the authors show that these structures, previously demonstrated to localize to discrete foci in cells, all share a common back-to-back architecture of hexagonally packed arrays. They propose that the (periplasmic, in membrane-bound arrays) ligand binding domains are thus minimally exposed to the cytosol. Through the finding that soluble domains of* E. coli *receptors form a similar structure, the authors argue that it may represent a default, low-energy configuration of chemoreceptor arrays rather than a specific evolutionary adaptation. However, the biological significance of this architecture is not discussed. Moreover, the interesting finding that the* R. sphaeroides *structure is curved is clouded by the lack of data from wild-type cells* – *the authors acknowledge this curvature may be due to the fluorescent proteins fusions*.

To clarify, we didn’t mean to propose that the ligand-binding domains are minimally exposed to the cytosol. Rather, we propose that this configuration is unlikely to inhibit access to signaling molecules as the receptor arrangement is much more porous than the outer membrane, which extracellular ligands need to cross in order to bind transmembrane arrays. To further clarify, we didn’t mean our text to suggest that this structure represents a default, non-evolved configuration. Rather, because we observe it both in cytoplasmic clusters and in soluble, normally transmembrane receptors from *E. coli,* we think this represents a fundamentally conserved architecture that has been evolved to optimize signaling properties. We have added electron tomography of untagged arrays, confirming that the curvature is not an artifact of fluorescent fusions.

*2) The initial description of receptor topology is somewhat confusing. Would a diagram be helpful? Some detailed information about the domains may not be necessary*.

We have added a diagram of the receptor topology in both membrane-bound and soluble chemoreceptor arrays in the new Figure 1. We have also taken out the overly detailed description of the kinase domains, as suggested.

*3) What new information does*
Figure 2
*add? It is helpful to have an orthogonal technique, but perhaps this could be combined with*
Figure 1.

The images of the embedded samples shown in the old Figure 2 (now Figure 3) portray a greater percentage of the overall cell, thereby lending better spatial localization information of the cytoplasmic clusters. Since the clusters are relatively small and have low contrast, reducing these panels to fit them into Figure 1 will make them too difficult to interpret. Additionally, we have added new data (Figure 8), showing that the curved architecture of the Rhodobacter cytoplasmic clusters is neither an artifact of fluorophore tagging nor a result of the ParA homolog PpfA. These experiments were done using the same technique as used for the sample shown in the old Figure 2 (now Figure 3) and are therefore directly comparable.

For these reasons, we would like to keep the figure as a separate figure. We have added the following sentence to the text: “It was difficult to identify the location of the structure in the cryosections, so we turned to a technique that would allow us to place the structure in the context of the whole cell. [...] The positions of the structures in the cells were consistent with FLM observations (47) strengthening our confidence in the identification of the cytoplasmic array.”

*4) The claim is made, based on*
Figure 2*, that the receptors are localized to the 1/4 and 3/4 positions. This should be backed up by quantification, and the mentioned similarity to light microscopy appropriately cited*.

Due to the necessarily low numbers of cells imaged, we don’t feel justified in making a quantitative claim here. Accordingly, we have scaled back our claim in the text and added references as suggested.

*5) One novel aspect of the* R. sphaeroides *chemoreceptor arrays is that they appear to be curved (though, as already mentioned, the exact nature of this curvature was not clear to me until later in the manuscript). The authors mention that the observed curvature may be due to the fluorescent protein tag. Imaging WT cells would obviously resolve this issue and would strengthen the manuscript considerably. While data they present from other species suggests that the formation of arrays can occur without a FP tag, the possibility that array formation is mediated by this tag remains*.

We have added electron tomography of untagged arrays, confirming that the array structure we observe is not an artifact of fluorescent fusions.

*6) What is the biological significance of arrays in general, and the cytoplasmic ones in particular? While large portions are devoted to structural details, the functional significance of these details is lacking*.

The full biological significance of cytoplasmic arrays remains unclear. As we discuss in the text, over half of all genomes sequenced of motile bacteria have more than one putative chemosensory pathway and many of these encode putative soluble chemoreceptors (55). In the case of *R. sphaeroides*, in which the cytoplasmic cluster is essential for normal chemotaxis, it has been speculated that cytoplasmic MCPs modulate chemotactic response based on the current metabolic state of the cell (4; 37), while the inputs of cytoplasmic receptors in other organisms such as *Vibrio cholerae* are still unclear. The results shown here and elsewhere reveal that the highly conserved 12 nm hexagonal array of receptor-trimers-of-dimers is a characteristic of both cytoplasmic and membrane-bound chemoreceptor arrays, and thus is likely key to the high sensitivity, wide dynamic range, cooperativity, and feedback control of this system (9; 26; 15; 42; 43; 27; 11). We have added this to the text.